# The Status of Pet Rabbit Breeding and Online Sales in the UK: A Glimpse into an Otherwise Elusive Industry

**DOI:** 10.3390/ani8110199

**Published:** 2018-11-06

**Authors:** Emma M. Gosling, Jorge A. Vázquez-Diosdado, Naomi D. Harvey

**Affiliations:** 1Politics and Society, Faculty of Humanities and Social Sciences, University of Winchester, Sparkford Road, Winchester SO22 4NR, Hampshire, UK; 2School of Veterinary Medicine and Science, The University of Nottingham, Nottingham LE12 5RD, Leicestershire, UK; svzjv@exmail.nottingham.ac.uk (J.A.V.-D.); Naomi.Harvey@nottingham.ac.uk (N.D.H.)

**Keywords:** rabbit, welfare, breeding, housing, husbandry, licence, legislation

## Abstract

**Simple Summary:**

Very little is known about where our pet rabbits come from: Who the breeders are, how good/or bad the conditions are that breeding rabbits are kept in, or whether breeders are being monitored by local authorities. This study aimed to bring to light information on breeding rabbits and breeders in the UK. Several methods of data collection were used combining data from online sales adverts, with a breeder survey and a council freedom of information request. From 3446 online rabbit sale adverts we found 94.5% of adverts were from England and only 1% of breeders were licenced. Out of 33 breeders surveyed, 51.5% provided smaller housing than recommended and housed most rabbits singly, against recommendations, and males were most likely to be housed singly, in too small conditions. However, most provided toys and a diet compliant with recommended guidelines. The most commonly sold/bred rabbits were breeds with flat-faces, which can cause significant health and well-being problems. A freedom of information request sent to 10% of UK councils revealed inconsistency in licensing conditions and confusion about eligibility. Without appropriate guidelines for housing and husbandry and regulation, rabbits within the pet rabbit breeding industry are at risk of compromised welfare.

**Abstract:**

Conditions of pet rabbit breeding colonies and breeder practices are undocumented and very little is known about the pet rabbit sales market. Here, multiple methods were employed to investigate this sector of the UK pet industry. A freedom of information request sent to 10% of councils revealed confusion and inconsistency in licensing conditions. Data from 1-month of online sale adverts (3446) identified 646 self-declared breeders, of which 1.08% were licensed. Further, despite veterinary advice to vaccinate rabbits from five weeks, only 16.7% rabbits were vaccinated and 9.2% of adult rabbits were neutered. Thirty-three breeders completed a questionnaire of which 51.5% provided smaller housing than recommended, the majority housed rabbits singly and bucks were identified as most at risk of compromised welfare. However, most breeders provided enrichment and gave a diet compliant with recommended guidelines. Mini-lops and Netherland dwarfs were the most commonly sold breeds, both of which are brachycephalic, which can compromise their health and wellbeing. From sales data extrapolation, we estimate that 254,804 rabbits are purposefully bred for the UK online pet sales market each year. This data is the first of its kind and highlights welfare concerns within the pet rabbit breeding sector, which is unregulated and difficult to access.

## 1. Introduction

Rabbits are the third most popular pet in the UK, with an estimated 1 to 1.5 million population [1,2]. However, very little is known about the pet rabbit population, even apparently basic things, such as which breeds are most common, are not recorded anywhere. Research indicates that in many cases the basic welfare needs of pet rabbits are not being met [3], but there is a complete absence of published data on the welfare of rabbits kept for breeding for the pet trade. To ensure provision of good welfare for any animal, the basic needs of the animal must be met through its housing and management conditions, both behaviourally and physically [4]. In many countries such basic needs are protected by law, however, if charities or governments are to put interventions in place to try and safeguard the welfare of animals, such as licensing, they must first know more about the animal population, for example how many there are and where they are located geographically. Whilst there are often official bodies in place and data available for farmed and laboratory rabbits, to our knowledge, there is no data available regarding the number of rabbit breeders, the conditions they are housed in or numbers bred each year for the rabbit pet trade in any country.

No specific guidelines exist for the housing and husbandry of rabbits bred for the UK pet trade. The British Rabbit Council (https://thebritishrabbitcouncil.org/) provides housing advice, but this does not include detailed guidance on meeting the welfare needs of a rabbit breeding colony. Therefore, breeders must refer to guidelines for pet rabbits given by the Welsh and Scottish governments [5,6], the Rabbit Welfare Association and Fund (RWAF) guidelines [7] or guidelines given for laboratory rabbits [8] or farmed rabbits [9]. Guidelines given by the RWAF are based on optimum welfare for rabbits, aiming to allow no deprivation to the welfare of the animals, and although not legally enforceable many elements are reflected in the enforceable Welsh and Scottish government guidelines for pet rabbits [5,6,7]. Guidelines for laboratory and farmed rabbits specify requirements that are less than those detailed for pet rabbits and focus not solely on rabbit welfare, but on efficiently performing scientific and farming procedures while administering the minimum amount of suffering, lasting harm, pain or distress to rabbits [8,9]. Despite the evidence expressing species specific welfare needs for captive rabbits (i.e., sufficient space and social interactions), the five welfare needs detailed in Section 9 of the Animal Welfare Act 2006 [10] is the only way in which UK legislation addresses, generally, the welfare needs of pet rabbits [11].

The five needs as defined in the Animal Welfare Act state that people have a duty of care to take steps towards meeting the following welfare needs of their animals: (1) A suitable environment to live in; (2) a suitable diet; (3) to be able to express normal behaviour; (4) to be housed with, or apart, from other animals; (5) to be protected from pain, suffering, injury and disease [10]. However, the needs of a breeding colony may be different from that of an animal kept as a pet. Research into rabbit needs has enabled the formulation of guidelines with legislation for keeping laboratory rabbits. The Code of Practice for the Housing and Care of Animals Bred, Supplied or Used for Scientific Purposes [8] stipulates minimum size and furnishing requirements for enclosures defining separate needs for breeding versus non-breeding colonies, as well as differing access to social partners and that breeding rabbit does need to be allowed to remove themselves from kits based upon evidence of wild doe behaviour [11]. Similar, but less extensive guidelines exist for farmed rabbits [9], but none for pet rabbits or pet rabbit breeding colonies.

Amongst pet rabbits, targeted surveys have revealed numerous welfare issues associated with pet rabbit housing and husbandry. According to the 2018 PDSA (People’s Dispensary for Sick Animals) Animal Wellbeing (PAW) report, which surveyed over 400 rabbit owners, most pet rabbits were purchased from pet shops or garden centres (36%), 17% were acquired from rescue/re-homing centres, and 15% of rabbits were from a friend, neighbour or family member [1]. The remaining 32% of rabbits are likely to be procured online, directly from breeders or as strays. Furthermore, 35% of rabbits lived in inadequate housing, 56% lived alone despite being a social species that require conspecific companionship, 44% had not been neutered, 50% had not been vaccinated and 32% had not been registered with a vet [1]. Extensive awareness campaigns regarding the need for rabbits’ diet to be comprised predominantly of hay and to avoid damaging rabbit ‘muesli’ have improved the diets of many pet rabbits seeing a reduction in muesli use from 49% of rabbits in 2011 to 20% in 2018 [1]. However, factors such as inadequate housing, lack of conspecific companionship and lack of preventative health care, remain prevalent amongst pet rabbits. Furthermore, access to exercise areas is often unpredictable. Many pet rabbits are kept in traditional hutches with separate exercise areas; of which one study reported only 23.5% of rabbits from 1254 owners were allowed continual access to their exercise areas [3]. Inappropriate or erratic access to exercise areas can have a negative impact the health and welfare of rabbits. Rabbits are naturally crepuscular and therefore allowing them access to runs at times when they are naturally most active is beneficial [3]. Continuous access also provides the rabbits with choice of where they want to be and providing some element of choice for an animal over its environment is a crucial component of good welfare [12,13,14,15]. With such issues being present within the pet rabbit population it is likely that welfare issues exist within rabbits kept for breeding purposes also, and with little to no information about this sector there is a need for investigation.

Two recent papers have demonstrated the utility of collecting data from online pet sale adverts for shedding light on pet relinquishments [16,17], yet to date there is no published data using this method to examine sales of rabbits sold by breeders. Some rabbit breeders are eligible for licensing by their local council under the Pet Shop Licence if selling direct to the public. Pet shop licences apply to any building deemed a ‘pet shop’, which can include home dwellings, however licences need not be issued to any person selling offspring from their own pet or pedigree animals [18]. As part of the licence, premises are inspected to ensure suitability of accommodation for the animals being bred and that their needs are being met. Licences are valid for one year and need to be renewed annually via application. The absence of consistent welfare guidance or enforceable legislation for pet rabbit housing within England is concerning as this indicates that owners, breeders and potentially council representatives are likely to be unaware of how to fully meet welfare needs of the rabbits in question.

The aim of this study is to fill an important gap in knowledge about the conditions of rabbits in the pet breeding trade and identify any specific welfare concerns. The overall aim of the study will be met by the following objectives:To identify the most common breeds, location of breeders, vaccination/neuter status at point of sale and self-declared licence information from data gathered from online sale adverts.To establish the housing and management conditions of pet rabbit breeding colonies via online questionnaire.To establish the consistency of licensing and welfare assessment of UK pet rabbit breeding colonies by local authorities.

As so little is known about the housing conditions of breeding rabbits, results will be compared to sources of legislation and guidelines for laboratory, farmed and pet rabbits. This study will provide important preliminary data on an opaque sector within the pet industry, the results of which would be relevant specifically to pet rabbit welfare in the UK, but will also highlight issues of relevance in any country where rabbits are kept as pets. Such work could lead towards further research and the formation of guidelines for the housing and care of pet rabbit breeding colonies, which could have a significant impact on rabbit welfare.

## 2. Materials and Methods

This research has been approved by the University of Winchester and the University of Nottingham ethics boards. The rabbit breeder questionnaire was anonymous and positive consent to participate was gained from all questionnaire respondents.

### 2.1. Council Pet Shop Licenses

A random sample of 40 councils (10%) in England, Scotland and Wales was selected from a total of all 407 councils via a random number generator from a list on the Local Government Information Unit [19]. The geographic distribution of the council areas was checked to ensure that the sample was evenly distributed. Councils from Northern Ireland were mitigated due to differences in pet shop licensing and bodies governing licensing (DAERA). A freedom of information (FOI) request was submitted to each council selected via email or online form (see Appendix A for dates). Information requested included: The number of new applications from rabbit breeders received in 2016, the number of new applications approved in 2016, whether the establishments applying for the licence were visited, the procedure for approving a pet shop licence application from a rabbit breeder and councils were asked to provide anonymised copies of all applications they had received. Replies from councils were put into an Excel spreadsheet and any documents sent were saved (see Appendix A). Councils who did not reply within the statutory 20 working days were contacted again two weeks after the cut-off date of 12th June 2017.

A second FOI request was sent to the same 10% of councils on 25th May 2018 to gather the same data as above for the year of 2017 and to include the number of renewal applications received for rabbit breeders in 2016 and 2017, again councils were re-contacted two weeks after the cut-off date of 20th June 2018.

### 2.2. Rabbits Sold Online

To estimate how many rabbit breeders are currently active within the UK and gather details on the rabbits being sold all adverts posted on the Pets4Homes website (www.pets4homes.co.uk) in a 1-month period (26th June 2018–26th July 2018) were collected and analysed. Pets4Homes was chosen to collect data from as it is the largest pet only classified advert website in mainland UK (England, Scotland and Wales). Data collected for each advert included: Location (town and county), breed, seller username, advertiser type (private seller or breeder/frequent advertiser), age of rabbits for sale, whether the advertiser deemed them ready for sale, whether the breeder was council licensed, and microchip, neutering and vaccination status. Data was collected via web-scraping using the *rvest* package [20] in RStudio (RStudio Team 2015, RStudio: Integrated Development for R. RStudio, Inc., Boston, MA URL http://www.rstudio.com/) and stored in an Excel spreadsheet for analysis.

### 2.3. Rabbit Breeder Questionnaire

To acquire information from UK rabbit breeders a questionnaire was developed based upon content of the Code of Practice for the Housing and Care of Animals Bred, Supplied or Used for Scientific Purposes [8] that details minimum enclosure sizes for does and their litters, pet shop license application forms and the five welfare needs to include questions relevant to minimum welfare standards. Questions collected information on; housing types (i.e., location, size, access to cover/hiding places, social interactions) and management practices (i.e., access to grazing, diet and enrichment). Questions were a mixture of free-text and categorical options (see Appendix B for the full questionnaire).

A Facebook page was created for the research project, this page was used to disseminate the breeder questionnaire to the general public. The questionnaire was disseminated via the social media sites Facebook and Twitter on 30 August 2017. Response numbers were monitored weekly and the questionnaire was reposted on other rabbit breeding group pages to attract breeders to complete the questionnaire. Where they could be identified, individual breeders were also contacted via their Facebook pages. The questionnaire was closed for responses on 31 January 2018. To increase sample size the questionnaire was reopened for responses on 26 April 2018 to 30 June 2018.

To widen the reach of the questionnaire, the RWAF was contacted to ask if they held a list of rabbit breeders, however they did not. The British Rabbit Council (BRC) was contacted on 24 June 2017 to ask if they possessed any guidelines on housing for breeding colonies of rabbits sold to the pet trade. They responded that they held no such information. The questionnaire was also sent via email to 12 individual breeders on the BRC directory whose email addresses were listed and to breeders whose contact details were found on the Pets4Homes website (www.pets4homes.co.uk). Only one pet sale website was used to avoid sending multiple emails to the same breeder. In order for the questionnaire to reach the targeted responders, those contacted from the pet sale website were those specifically listed as a ‘breeder’.

#### Data Analysis

All data was stored in Excel spreadsheet and each data set collected from the questionnaire was assigned a unique ID number. Questionnaire responses and council responses contained both qualitative and quantitative data. Where possible descriptive statistics were used for quantitative data. All free text answers were analysed via thematic analysis [21], by grouping similar ideas, phrases and words within the respondent answers to form themes. Themes created were then applied to other answers within the same question and numbers of breeders fitting each theme were counted.

## 3. Results

### 3.1. Council FOI Requests

Of the FOI requests sent to councils in 2017 requesting 2016 license information, 40/40 were returned representing 10.3% of all UK councils (Appendix A). One council did not hold the requested information and two reported they were not responsible for licensing. Out of the 37 valid responses, just one pet shop licence for rabbit breeding was applied for in 2016 at Amber Valley Borough Council, which was unsuccessful. Not one of the 37 councils responsible for licensing had granted a valid licence for any rabbit breeding establishment in 2016, nor had they received any licence renewal applications.

For the FOI requests sent in 2018 to the same councils covering 2017 licence information, 40/40 were returned. One council did not record whether pet shop licence applicant bred rabbits, two councils still reported that they were not responsible for licensing, one council (East Hampshire District Council) received and approved a new pet shop licence application from a rabbit breeder and received one renewal application, and Amber Valley Borough Council received a renewal application which was visited by an inspector; 36 of these 40 councils reported no licensed rabbit breeders in 2017. The online sales data collected in 2018 from 26th June to 17th July revealed seven breeders who said they were licensed, one each at two of the councils surveyed by the FOI request: Amber Valley Borough Council and East Hampshire District Council; and one each at five additional councils: Manchester City Council, North Lanarkshire Council, Northumberland County Council, South Lakeland District Council and Wakefield Metropolitan District Council.

#### Licensing Procedures

In response to the open question in the 2017 FOI request ‘Could you please detail your procedure for approving pet shop licenses for rabbit breeders that sell for the pet trade?’, one council stated that “sales made to the trade and not direct to the public do not require a pet shop licence”, another said that a licence was needed if selling animals from a commercial premises or via the internet, and a third council said that all pet shops selling rabbits needed to keep a register of where rabbits were purchased i.e., suppliers and where they were sold. With regards to licensing procedure, six (15%) used veterinary input to inform decisions on licence approval, 10 (25%) inspect the premises that the licence has been applied for, six (15%) used guidance of legislation to inform licensing, 13 (32.5%) stated they did not have a specific procedure and six (15%) considered the question not applicable (answers are not mutually exclusive).

### 3.2. Online Sales

In the four weeks from 26th June 2018 to 17th July 2018, 3446 unique adverts on Pets4Homes were identified for rabbits for sale. Amongst these, were 646 self-declared breeders posting 1910 adverts, of which only seven breeders, just 1.08%, were council licensed. A total of 173 breeders were within the council areas where the FOI requests were sent (see Appendix C
Table A1). The remaining adverts were from private sellers (1529 adverts) and rescue centres (7 adverts). Of all adverts, the majority were from England (94.5%), whilst 3.9% were from Wales and 1.6% were from Scotland.

The most common breeds advertised were Mini-lop (39.3%), mixed breed (14.5%), Netherland Dwarf (13.8%) and Lionhead (9.7%) (see Appendix A for a full list of breeds). A total of 2548 (74%) adverts were for recently bred rabbit kits aged 16 weeks and under, of which 59.8% came from self-declared breeders and 40.1% came from private sellers and 0.1% came from rescue centres. Of the 898 advertisements posted for rabbits aged over 16 weeks for which neuter status was given (mean age of 14 months), just 83 were neutered (9.2%) and 59 (6.6%) were micro chipped. The adverts for adult rabbits identified here could be considered to represent a relinquished population, which could explain the low levels of neutering, as has been shown elsewhere [22].

Despite veterinary advice to vaccinate rabbits from five weeks of age [23] only 16.7% (459) of the 2739 adverts that detailed vaccination status said the rabbits had up to date vaccinations. As per the website rules, no rabbits were advertised as ready to be sold of less than eight weeks of age.

### 3.3. Rabbit Breeder Questionnaire

A total of 33 responses were received from the online rabbit breeder questionnaire. Overall, breeders had between 6–10 does and 4–6 bucks, and 33% bred multiple types of rabbits. The most common breed was the Mini-lop at 63.6%, followed by Netherland Dwarf and Lionhead 18.2%, Dutch and Mini Rex 9.1%, French lop and Lion Lop 6.1%. The least common breeds were Giants, Belgian Hares, Dwarf Lop, Mini Plush, Mini Dutch, Silver Fox and New Zealand White 3% (Appendix A). Two breeders gave vague descriptions of their breed types: ‘Lops’ and ‘all types’.

#### 3.3.1. Housing and Husbandry

Singly housed rabbits had the lowest median housing size, with bucks having the smallest median housing size overall (Figure 1). All group housed rabbits, with two exceptions, who both housed pairs not groups of rabbits, had a floor area that exceeded the minimum recommended floor area (7500 cm^2^) for a group of three laboratory rabbits over 10 weeks of age weighing 2.5–3 kg [8].

Adult bucks and does have a recommended floor area of 7500 cm^2^ in farmed rabbits [9], 10 breeders did not meet this minimum requirement for their adult rabbits. When considering the minimum requirements for housing pet rabbits (11,148 cm^2^), 17 breeders did not meet these for some or all of their rabbits. Four breeders did not indicate housing height for all rabbits, and 21 breeders (63.6%) provided housing height less than the 75 cm recommended by the Scottish Government for pet rabbits [6]. One breeder provided housing that was below the minimum recommended height of 45 cm for laboratory and farmed rabbit housing [8,9]. All housing was cleaned at least weekly by breeders. 

Bucks were housed singly by breeders more often than any other type of rabbit (76.7%), and group housing was rare with only three breeders housing bucks in pairs and three housing bucks in groups, two of which were in a bachelor colony.

For does without kits, 49.5% housed them singly, 24.2% housed some singly and in pairs, 9.1% housed them in pairs, 9.1% housed them in groups and 6.1% did not specify how does were housed. For does with kits, 33% (11) housed does with kits in groups with 67% housing them singly. 

#### 3.3.2. Breeding Does

Breeding does had a mean breeding lifetime of 24.4 months (minimum four months, maximum 60 months) with a mean starting age of 7.8 months (Figure 2) and an average of 1.8 litters per year (minimum 1, maximum 5). Two breeders did not specify an age when the does were retired; just saying that they were retired when ready to retire or when they no longer had litters. Does were not given areas to get away from their kits by 36.4% (12) breeders. Litters per year varied with the shortest time left between litters being two months per doe, the mean being six months and the maximum being 12 months. Retired does were either always kept by their breeders (24.2%), always re-homed or sold (21.2%) or sometimes kept, sometimes re-homed (51.6%) whilst 3% did not specify whether they were kept or re-homed.

#### 3.3.3. Kits—Handling and Age Sold

All but one breeder handled their kits with 85% (28) starting handling between birth and leaving the nest. Out of the breeders who handled their kits, all except one handled kits at least daily (91%). Of all breeders, one took into consideration whether the doe was happy with handling before handling the kits.

Breeders sold kits directly to the general public (75.8%), pet shops (18%), as show animals (3%) and to other breeders (15%). Three breeders (9%) indicated that they sold kits online. Two breeders did not specify who they sold their kits to (6.1%) and three (9.1%) did not specify the age at which they were sold. The average minimum age when kits were sold was 8.5 weeks (minimum five weeks, maximum twelve weeks) with two breeders selling their kits before the recommended minimum threshold of eight weeks [24].

#### 3.3.4. Diet

All breeders indicated that they gave access to fresh grass for all their rabbits, however when asked to detail their feeding regime, only four breeders included fresh grass. In compliance with the RWAF guidelines (85% hay, 10% greens and 5% pellets [25]); appropriate amounts of hay (body sized), a small number of pellets/pellets once daily and greens three or more times a week was given to 45.5% of bucks, 45.5% of does without kits, 33.3% of does with kits and 36.3% of juveniles (Figure 3). Those who gave mostly hay with or without pelleted food and/or greens were considered to be in moderate compliance with RWAF guidelines and accounted for 42.4% of bucks, 45.5% of does without kits, 57.6% of does with kits and 51.5% of juveniles, and those who gave a poor diet; not including hay, and/or feeding of muesli and/or bread accounted for 12.1% of bucks and juveniles and 9.1% of does (Figure 3).

#### 3.3.5. Exercise

Rabbits were allowed access to runs by 31/33 breeders (93.9%). One breeder specified that rabbits had access to runs several times daily but did not give run measurements. Two breeders specified “free roam” or “large runs” but did not give specific measurements. Access to runs was variable dependent upon doe status (litter or no litter), age and sex but daily access to runs was the most common (Figure 4). Only a few breeders allowed rabbits continuous access to a run (9% for bucks and 15% all others), or access to a run several times daily (12% for all types). Infrequent or no access to runs was only apparent in two breeders. Breeders whose rabbits had infrequent access to runs gave reasons for reduced access including; the rabbits were rotated on a daily basis, and “they have their large 3ft hutches”. Run sizes were specified by 28/33 breeders (84.8%) of which 70% of runs were smaller than the RWAF recommended minimum for a pair of rabbits (44,593 cm^2^).

#### 3.3.6. Enrichment

With the exception of one, all breeders provided all rabbits with environmental enrichment (97%). Enrichment varied, with balls being the most common enrichment provided by 58% of breeders. Chew toys were the next most common enrichment given by 36.4% of breeders, followed by cardboards tubes (30%) and boxes and hay racks (24.2%). Only twelve breeders (36.4%) provided hiding places in the form of boxes or tunnels. Foraging behaviour was encouraged by 45.5% of breeders by providing treat filled boxes/balls or hay filled logs or tubes.

## 4. Discussion

The aim of this study was to fill in a gap in knowledge about the conditions of rabbits in the pet breeding trade and the demographics of the UK rabbit sales market. The following objectives were fulfilled in order to meet this aim: An online questionnaire investigating the housing and husbandry of UK breeding rabbits was distributed via social media; results from 33 breeders were compared to existing guidelines/legislation for laboratory, farm and pet rabbits. The current status of licensing of rabbit breeders in 10.3% of UK (England, Scotland and Wales) councils was established. One months’ worth of data from online advertisements was collected from the Pets4Homes website detailing the rabbits for sale and the status of these rabbits including; age, breed, location, council license status, neuter and vaccination status and seller type. This information can be used to estimate the newly bred rabbit population in the UK, the most common breed types, age at which kits are sold and location of breeders with relation to the council areas sent an FOI request. Information such as this is important to collect as the status of rabbits being bred for sale is an unknown area and this data provides the only known overview of pet rabbit breeding. It also provides a comparison to the council data for how many breeders are currently licensed compared to how many self-declared breeders were actively selling rabbit kits.

A recent paper from Neville et al. [17] identified 224,563 adverts for adult rabbits (data for juveniles was not presented) sold online over a three-year period between 2014–2016 in the UK across six sales platforms, including Pets4Homes. Based upon our online sales data from Pets4Homes, 74% of adverts were for recently bred rabbit kits aged 16 weeks and under, of which 59.8% came from declared breeders. Extrapolating from these figures (assuming the data from the adult rabbits in Neville et al. represented 26% of all adverts), we estimate that there could be as many as 213,046 adverts for recently bred rabbits placed online each year, of which 127,402 could be coming from self-declared breeders. It’s important to note that one advert does not equal one rabbit, with many adverts containing reference to multiple rabbits from a single litter the actual figure for young rabbits bred by breeders and sold online each year could be double this figure conservatively speaking; 254,804. With online sales estimated to represent just 6% of owner purchases according to the 2018 PDSA survey [1], this is likely to be an underestimate compared to the actual number of rabbit kits sold each year for the pet trade in the UK. It will be important for future studies to continue to collect data of this kind in the hope of refining this extrapolation. With such potentially vast numbers of animals involved, it is vital that safeguards are put in place for the wellbeing of the kits and the breeding colonies. Online sales in particular represent an increasing welfare problem; data from an American study indicates that breeders of dogs advertised online were less likely to screen their animals for heritable diseases and were less aware of health and welfare issues than dog breeders who did not sell their puppies online [27]. Further, to our knowledge, rabbits sold online are unlikely to be sold with any care advice or guidance other than spoken advice from the breeder, such as may be provided when they are purchased from pet shops.

To our knowledge, there were no published accounts identifying the most common rabbit breeds in the UK prior to this work. This may be because it has been deemed information that is not important, but with the rise of welfare problems due to breeding e.g., brachycephalism (shortened skulls) in dogs [28], cats, and rabbits [29,30], it is important to establish how relevant these welfare issues may be in the rabbit population. Both the questionnaire and Pets4Homes data highlight the Mini-lop (63.6% and 39.3% respectively) as the most common breed with the next two most common pure breeds being the Netherland Dwarf (18.2% and 13.8% respectively) and lion head (18.2% and 9.7% respectively). All of these breeds have brachycephalic skull shapes, to a greater or lesser extent. There are severe welfare problems associated with brachycephalism in rabbits most of which stem from dental disease, including brachygnathism (shortened jaw) and incisor malocclusion [31]. In severe cases this misalignment of the teeth causes overgrowth of teeth and eventually lacerations of the mouth, abscesses and potentially death in addition to misshapen tear ducts cause tearing or pus running onto the face [29,31]. Incisor malocclusion is reported to be common in dwarf breeds, including the Netherland Dwarf, with some breeders known to cull affected kits early in life [31]. Recent research in Taiwan shows brachycephalic rabbit breeds are 3.12 times more likely to suffer from dental diseases than non-brachycephalic breeds [32], further, rabbits with lop ears commonly develop middle ear infections [33]. With these prominent welfare issues, the popularity and excessive breeding of mini lops and other brachycephalic breeds is a considerable welfare concern.

The Scottish and Welsh guidelines on rabbit welfare from 2018 and 2009, respectively, provide guidance consistent with each other and the RWAF on suitable rabbit housing (minimum floor area 11,148 cm^2^, minimum height 75 cm) [5,6,7]. Such guidelines are for pet rabbits only and no such guidelines exist for pet rabbits kept in England despite the fact that 94.5% of rabbits sold on Pets4Homes originated in England.

In our breeder questionnaire the majority of breeding rabbits (49.5% to 76.7% depending on rabbit type) were housed singly, with just 10 breeders housing rabbits in pairs. Singly housed adult rabbits had the lowest average floor area, with bucks having the smallest overall housing. In many cases single bucks’ housing was smaller than single does’ and six breeders (18%) who housed their rabbits singly had floor areas below the minimum recommended for laboratory rabbits. With regards to height requirements for housing 3% of breeders housing was below laboratory requirements and 61% was below pet requirements; this limits behavioural expression, which can result in abnormal and/or stereotypic behaviour [34]. Singly housed rabbits in small pens often show limited behaviour compared to group housed rabbits [25] and have higher incidence of abnormal behaviour [35], as well as reduced locomotor and exploratory activity [34]. Reduced locomotion over a long time period results in physical malformations e.g., spinal deformity; thus causing pain and suffering, reducing the welfare of the rabbits [35].

In addition to being housed in the smallest areas, bucks were housed singly by breeders more often than any other type of rabbit (76.7%). There is no ecological reason to justify housing bucks in smaller areas or singly more than does, yet our finding supports that of the 2018 PDSA report, which found that owners of male rabbits are significantly more likely to house them alone (67%) than owners of female rabbits (35%) [1]. Taken together these findings imply that bucks may be the most at risk of having compromised welfare.

Although some breeders may purchase specified breeding rabbit housing, it is likely that many breeders will purchase commercially available pet rabbit housing. The ability to purchase hutches and runs that do not meet minimum welfare standards needs to be addressed. Minimum requirements for optimum welfare, such as those stipulated by the Welsh and Scottish governments should be implemented across the UK and accommodation that does not meet these requirements should not, in our opinion, be sold. Further, there is a need to develop guidance specifically for rabbits kept for breeding, which is currently overlooked by all guidance excepting that given for rabbits used for scientific and meat purposes.

Dental disease is the most common problem described by vets in UK domestic rabbits, which includes breeding rabbits. Rabbits’ teeth grow continually, thus the correct diet high in fibre from hay or grass needs to be fed in order to wear the teeth down adequately and prevent dental disease. Abnormal wear of the teeth leads to malocclusion, which if unnoticed can result in serious health and welfare issues, including death [36]. Here, rabbits fed a diet closest to the recommended guidelines from the RWAF [26] were bucks, and does without kits (45.5% of breeders), with does with kits and juveniles matching the recommended diet in the least cases. The BRC provides specific guidance on juvenile rabbits and lactating does: “Growing rabbits should be expected to eat up to twice the amount consumed by an adult and lactating does three times the amount. In lactating does, energy needs often exceed capacity for food intake and weight will be lost” [37]. Although not all breeders surveyed here fed the recommended diet for does with kits and juvenile rabbits, 42% increased the food intake of their does with kits and 48% increased food intake for juveniles, and the majority of rabbits (89%) received diets high in hay. This is positive as it indicates that many breeders are conscious of the nutritional needs of their rabbits and the increased need for growing and lactating animals. However, there is still room for improvement that could be achieved with proper guidance on dietary requirements for breeding rabbits.

Almost all breeders surveyed here provided enrichment for their animals, and the majority of enrichment was food orientated to promote foraging behaviour indicating that breeders have some knowledge of their animals’ welfare. However, ecological relevance has been shown to be of great importance in selecting enrichment material for captive animals [38]. Most breeders surveyed here provided balls, which are not an ecologically relevant source of enrichment, yet 63.6% of breeders failed to provide ecologically relevant enrichment in the form of hiding places, such as boxes or tunnels. Such a statistic is concerning as inadequate access to sheltered areas can increase stress of pet rabbits due to them having a natural fearfulness of open spaces, and inability to hide away from predators or intense sunlight can be perceived as a welfare issue [3].

In order for authorities to be able to enforce the animal welfare laws outlined in the Animal Welfare Act (2006) [10] they must first be able to identify the animals in need of protection and the people responsible for them. Licensing of animal breeding and sales is the best way to safeguard the welfare of breeding animals as it requires breeders to be traceable, and to meet minimum welfare requirements. Establishments where rabbits are bred for commercial purposes or as a business are required to be licensed [39] this includes sale of animals over the internet by businesses [40] although this is not always clear. Pet shop licenses are not required for the sale of: “[A]nimals that are offspring of your own pet” or “pedigree animals that you’ve bred” [18]. Interpretation of the aforementioned statements may differ between individuals and bodies (for example some councils we surveyed did not feel an approval procedure was applicable), specifically with regards to what constitutes a pedigree rabbit, meaning that there can be confusion as to whether an establishment requires licensing or not. Further to this, the pet shop licensing information provided by the UK government does not detail what minimum qualities constitute a pet shop; the statement given is simply “a pet shop is any building where animals are sold as pets, including your own home” [18].

The random sample of councils contacted here received one license application for rabbit breeders in 2016 (Amber Valley District Council) and one license application in 2017 that was inspected (East Hampshire District). A total of seven self-declared breeders advertising on Pets4Homes in June-July of 2018 reported to hold a council license, of which one was in Amber Valley District Council and one East Hampshire District suggesting that these councils renewed the licenses in 2018 also. Considering the 2018 estimate of seven breeders being licensed in the UK would mean that only 1.08% of self-declared breeders selling directly to the public were actually licensed to do so. These results suggest that the majority of the current population of more than one million UK pet rabbits must be bred by unlicensed, and therefore unregulated, breeders. This is a problem, as it means that there are no animal welfare safeguards in place for 98.92% of the UK rabbit breeding population (such as site inspections and maintained records), and no way to know where the animals are or to make irresponsible breeders accountable. For this reason the RWAF have launched the Capone Campaign (https://rabbitwelfare.co.uk/campaigns/capone-campaign/) where the charity have committed resources to monitor online rabbit sales and report irresponsible unlicensed sellers directly to councils and the RSPCA in order to make authorities aware of the problems caused by unlicensed pet sales [41].

On 1st October 2018 new regulations came into force: The Animal Welfare (Licensing of Activities Involving Animals) (England) Regulations 2018 [24], which gives detailed guidance on local authority licensing and conditions that must be met by licensing bodies and people applying for a licence, as well as stipulating that all commercial sales require a licence, including those taking place online. While this new guidance offers elaboration on the five welfare needs and improves the Pet Animals Act 1951, as well as mentioning species specific needs, it still does not contain specific guidance for breeding rabbits or pet rabbits, and it fails to define what constitutes a commercial sale. More guidelines are needed with rabbit specific conditions to improve welfare of breeding rabbits, although these new regulations are undoubtedly a step in the right direction.

## 5. Conclusions

Our results suggest that there are prevalent welfare issues within the UK pet rabbit breeding sector. Although only a small percentage of rabbit breeders took part in this study, significant diversity was seen in housing and husbandry conditions available to breeding rabbits. Housing and run size was variable, with 18.2% of bucks and 6.1% of does housed in restricted conditions smaller than the minimum recommended cage size for laboratory rabbits, and 33% of breeders housed rabbits in hutches smaller than the minimum recommended housing size for pet rabbits. Furthermore, social requirements of breeding rabbits were seldom met by UK rabbit breeders with the majority of bucks and does being housed singly. Diet and enrichment were two areas that presented the least welfare concerns from rabbit breeders; the majority of breeders showed knowledge of the need for hay in rabbits’ diet, however just over half of breeders failed to increase nutritional provisions to meet the needs of pregnant and lactating does, meaning there is room for improvement. Enrichment was provided by all but one breeder; however it is unclear whether breeders provided the right types of enrichment to fulfil their rabbits normal and natural behaviour needs with the most common enrichment being a non-ecologically relevant ball and 63.6% of breeders failing to provide ecologically relevant enrichment in the form of hiding places.

Online sales data implies that most rabbit kits are sold un-vaccinated, despite veterinary advice to vaccinate from five weeks of age. Further, we have been able to extrapolate the first estimate of the rabbit size of the rabbit breeding market, with a possible 254,804 rabbits purposefully bred for the UK online pet sales market each year; the majority which appear to be bred in England. The most commonly sold rabbit breeds were identified as being Mini-lops and Netherland dwarfs; both breeds with brachycephalic conformations, which can have devastating health consequences.

The data collected here could be of value for organisations to implement interventions to safeguard the welfare of breeding rabbits. Council licensing could be used as a safeguard, through which the Animal Welfare Act (2006) [10] could be implement, however our results show that licensing is an area with multiple shortfalls when it comes to rabbit breeding. The current legislation is not applicable to the modern world of online sales and the responses received from the councils surveyed here suggest is it not clear enough to enable comprehension. The majority of breeders in the UK appear to be unlicensed, and are therefore untraceable and unaccountable for their animals’ welfare.

The variability in practices by breeders may be due in part to the distinct lack of guidance available to clearly detail standards for providing optimum welfare amongst rabbit breeding colonies. Therefore, if the housing and husbandry conditions, and thus welfare of breeding rabbits in the UK is going to improve it is paramount that consistent, standardised guidance is provided to rabbit breeders and that appropriate, clear legislation is created regarding eligibility for licensing.

## Figures and Tables

**Figure 1 animals-08-00199-f001:**
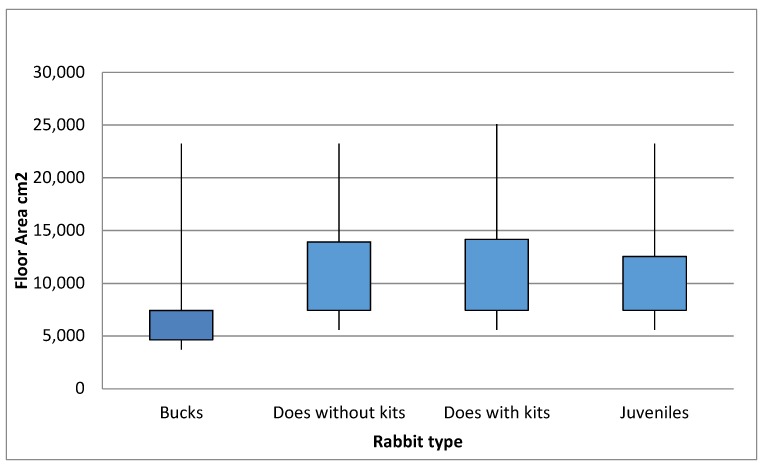
Boxplots showing the range, and interquartile range, of housing sizes provided for singly housed rabbits.

**Figure 2 animals-08-00199-f002:**
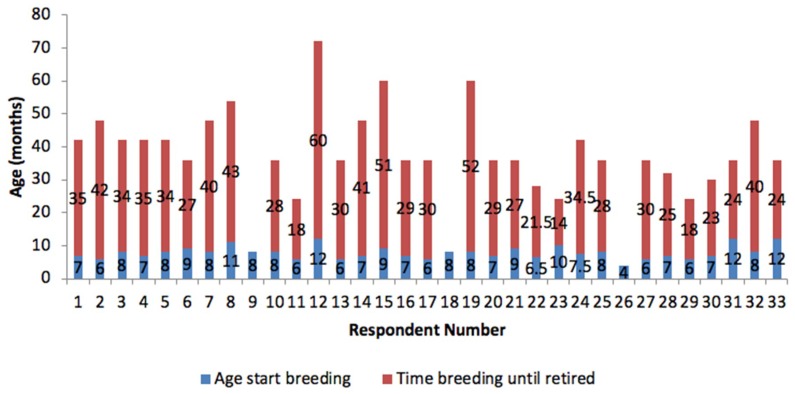
Breeding age and months until cessation of breeding for does shown per breeder.

**Figure 3 animals-08-00199-f003:**
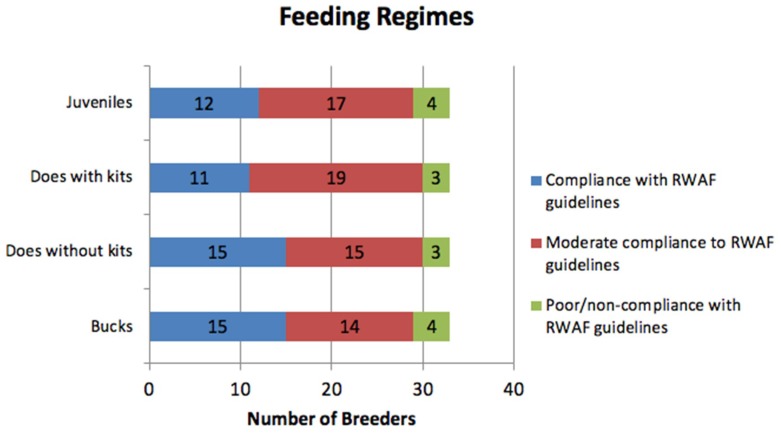
Compliance of feeding regimes with guidelines for diet provided by the Rabbit Welfare Association and Fund (RWAF) [26].

**Figure 4 animals-08-00199-f004:**
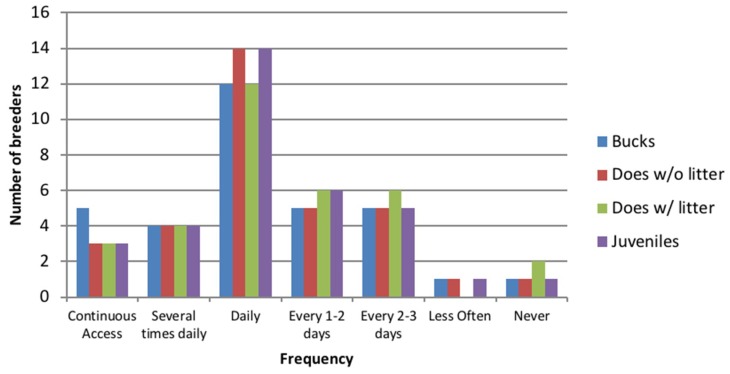
Access provided to runs by breeders for difference types of rabbit within the breeding colony.

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
