# Peer review of "The Status of Pet Rabbit Breeding and Online Sales in the UK: A Glimpse into an Otherwise Elusive Industry"

_animals, 2018, doi:10.3390/ani8110199_

Round 1

Reviewer 1 Report

I think this study is a really useful look at the pet rabbit breeding world in the UK, an area which is typically quite understudied. I would like to see a bit more said about the various welfare standards referenced by the authors, as the standards developed for laboratory rabbits could be very different from those developed, for example, by the RWAF. Could the authors speak briefly to the differences among those standards and under what conditions those standards were developed? I think this could help the reader to assess how rigorous those standards are, and whether they are significantly different.  In addition, can the authors reference the different ways that people in the UK get rabbits, ie via rescuers, breeders, etc.? I think it would be useful to note the other ways in which people are acquiring rabbits, as acquiring rabbits through rescuers is becoming a more popular method.

Author Response

Point 1: I would like to see a bit more said about the various welfare standards referenced by the authors, as the standards developed for laboratory rabbits could be very different from those developed, for example, by the RWAF. Could the authors speak briefly to the differences among those standards and under what conditions those standards were developed? I think this could help the reader to assess how rigorous those standards are, and whether they are significantly different.

Information has been added detailing the motivations of the guidelines referenced and how they are employed to help the reader better understand how they affect the rabbits cared for under the guidelines. 

Point 2: In addition, can the authors reference the different ways that people in the UK get rabbits, ie via rescuers, breeders, etc.? I think it would be useful to note the other ways in which people are acquiring rabbits, as acquiring rabbits through rescuers is becoming a more popular method.

Statistics included from the most recent PDSA PAW report detailing how pet rabbits were purchased.

Reviewer 2 Report

The authors present a unique set of data that make a limited but useful contribution to our understanding of how private breeders may contribute to or take away from animal welfare.  I have very few comments about the paper—most have to do with grammar or with the use of the word “popular” when the data being presented are what is most common.  To know about popularity or preference, one has to ask others.  What is most common is not the same as what is most popular.  The authors also repeat far too much of their results in the Discussion section of the paper, and the Conclusion section is repetitive to the Discussion section.

Minor comments are as follows:

Line 38  “…and inaccessible.” 
  If inaccessible, how did YOU assess it?  Change this word as it causes some confusion here.

Line 156:  “…and stored in an excel spreadsheet for analysis.” 
  Excel is a proper noun, and should be capitalized here, as indeed it is elsewhere in the manuscript.

Line 228:  “…The most popular breeds advertised were Mini-lop…”  The data that are being reported are on what was the most common breed reported by respondents to the authors’ survey.  To the best of my understanding, the authors did NOT ask breeders what the most popular rabbit breed was.  Rather, they asked what was the breed that they produced most.  What is most common is not necessarily the same as what is most popular.  Do not make more of these data than they allow.

Lines 242-243:  “…The most popular breed was the Mini-lop at 63.6%...The least popular breeds were Giants…”  Same problem here.

Lines 248-252:  “…All group housed rabbits, with the exception of breeders 8 and, for juveniles breeder 21, who both housed pairs not groups of rabbits, had a floor area that exceeded the minimum recommended floor area (187500px2) for a group of three laboratory rabbits over 10 weeks of age weighing 2.5-3kg [7]. “

This sentence is very hard for the reader to understand.  Who or what is “breeders 8?”  I think you mean respondent #8, but since you have not yet introduced this way of referring to particular subjects to the reader, it is completely confusing.  The same is true for “breeder 21.”  I had to read this sentence multiple times to try to understand it.  You could just say “with two exceptions” and leave the level of detail as to which breeders were the exceptions out of this sentence.

Lines 301-302:  “…those who gave a poor diet with no hay, muesli and bread accounted for 12.1% of bucks and juveniles and 9.1% of does (Figure 3). “

Again, this sentence is pretty confusing.  Do you mean a poor diet with no hay and only muesli and bread?  Or do you mean with no hay, no muesli, and no bread?  Or what?  Rearrange this so that it makes sense.

Lines 313-314:  “Reasons given for rabbits not allowed frequent access to runs “less often” or “never” were; rabbits are rotated daily (one breeder) and “they have their large 3ft hutches” (one breeder).”

I cannot understand this sentence as written.

Line 337: “…the most popular breed types …”  Again, what is most common is not the same as what is popular.

Lines 361-362:  “…Further, rabbits sold online are usually sold without any care advice or guidance such as may be provided when they are purchased from pet shops.”  Support this statement with data or make it clear that this is anecdotal or speculation.

Lines 367-369:  “…Both the questionnaire and Pets4Homes data highlight the Mini-lop (63.6% and 39.3% respectively) as the most popular breed with the next two most popular pure breeds being the Netherland Dwarf ..”  What is most common is not the same as what is most popular.  Do not make more of your data than they allow.

Lines 480-482:”… with 18.2% of bucks and 6.1% of does housed in restricted conditions smaller than the minimum recommended cage size for laboratory rabbits, and 33% of breeders housed rabbits in hutches smaller than the minimum recommended housing size for pet rabbits.”  The authors present these data as though they are a serious concern, but in all truth, these numbers are not very big, and thus most rabbits (>80% of bucks and 93% of does) are being housed in conditions larger than what is recommended for lab rabbits.  Or am I reading this incorrectly?  A change in tone would be appropriate here. 
The alarmism is not justified by the data presented.

Author Response

Point 1: Line 38  “…and inaccessible.” 
  If inaccessible, how did YOU assess it?  Change this word as it causes some confusion here.

Point 2: Line 156:  “…and stored in an excel spreadsheet for analysis.” 
  Excel is a proper noun, and should be capitalized here, as indeed it is elsewhere in the manuscript.

Line 38 and 156 have been changed using the reviewers suggestions.

Point 3: Line 228:  “…The most popular breeds advertised were Mini-lop…”  The data that are being reported are on what was the most common breed reported by respondents to the authors’ survey.  To the best of my understanding, the authors did NOT ask breeders what the most popular rabbit breed was.  Rather, they asked what was the breed that they produced most.  What is most common is not necessarily the same as what is most popular.  Do not make more of these data than they allow. 

Point 4: Lines 242-243:  “…The most popular breed was the Mini-lop at 63.6%...The least popular breeds were Giants…”  Same problem here.

All mentions of popularity of breed have been changed to highlight commonality of breeds.

Point 5: Lines 248-252:  “…All group housed rabbits, with the exception of breeders 8 and, for juveniles breeder 21, who both housed pairs not groups of rabbits, had a floor area that exceeded the minimum recommended floor area (187500px2) for a group of three laboratory rabbits over 10 weeks of age weighing 2.5-3kg [7]. “

This sentence is very hard for the reader to understand.  Who or what is “breeders 8?”  I think you mean respondent #8, but since you have not yet introduced this way of referring to particular subjects to the reader, it is completely confusing.  The same is true for “breeder 21.”  I had to read this sentence multiple times to try to understand it.  You could just say “with two exceptions” and leave the level of detail as to which breeders were the exceptions out of this sentence.

The suggested changes have been made to lines 248-252

Point 6: Lines 301-302:  “…those who gave a poor diet with no hay, muesli and bread accounted for 12.1% of bucks and juveniles and 9.1% of does (Figure 3). “

Again, this sentence is pretty confusing.  Do you mean a poor diet with no hay and only muesli and bread?  Or do you mean with no hay, no muesli, and no bread?  Or what?  Rearrange this so that it makes sense.

The sentence has been rearranged and elaborated on to make it clear, in this case, what constituted a poor diet. 

Point 7: Lines 313-314:  “Reasons given for rabbits not allowed frequent access to runs “less often” or “never” were; rabbits are rotated daily (one breeder) and “they have their large 3ft hutches” (one breeder).”

I cannot understand this sentence as written.

This sentence had been rearranged to hopefully make sense and provide clarification on meaning.

Point 8: Line 337: “…the most popular breed types …”  Again, what is most common is not the same as what is popular.

This has been addressed as suggested

Point 9: Lines 361-362:  “…Further, rabbits sold online are usually sold without any care advice or guidance such as may be provided when they are purchased from pet shops.”  Support this statement with data or make it clear that this is anecdotal or speculation.

Re-worded the sentence to avoid presenting it as a statement of fact

Point 10: Lines 367-369:  “…Both the questionnaire and Pets4Homes data highlight the 
 Mini-lop (63.6% and 39.3% respectively) as the most popular breed with the next two most popular pure breeds being the Netherland Dwarf ..”  What is most common is not the same as what is most popular.  Do not make more of your data than they allow.

This has been addressed as suggested

Point 11: Lines 480-482:”… with 18.2% of bucks and 6.1% of does housed in restricted conditions smaller than the minimum recommended cage size for laboratory rabbits, and 33% of breeders housed rabbits in hutches smaller than the minimum recommended housing size for pet rabbits.”  The authors present these data as though they are a serious concern, but in all truth, these numbers are not very big, and thus most rabbits (>80% of bucks and 93% of does) are being housed in conditions larger than what is recommended for lab rabbits.  Or am I reading this incorrectly?  A change in tone would be appropriate here. 
The alarmism is not justified by the data presented.

Although these percentages are only relatively small, the size of cages laboratory rabbit do not meet the suggested welfare requirements for good rabbit welfare as set forth by the RWAF. If rabbits spend the majority of their time in these very small cages it is likely they will suffer physical deformities which will cause pain and suffering (as mentioned in the introduction of the paper). Therefore as breeding rabbits are essentially classed as pets their rabbits offspring are intended to be sold as pets, keeping them in conditions equal to or less than laboratory rabbit conditions diminishes welfare significantly so I think the tone of the aforementioned presented data is justified.